# Historical Selection, Adaptation Signatures, and Ambiguity of Introgressions in Wheat

**DOI:** 10.3390/ijms24098390

**Published:** 2023-05-07

**Authors:** Demissew Sertse, Frank M. You, Valentyna Klymiuk, Jemanesh K. Haile, Amidou N’Diaye, Curtis J. Pozniak, Sylvie Cloutier, Sateesh Kagale

**Affiliations:** 1Aquatic and Crop Resource Development, National Research Council Canada, Saskatoon, SK S7N 0W9, Canada; 2Ottawa Research and Development Centre, Agriculture and Agri-Food Canada, Ottawa, ON K1A 0C6, Canada; frank.you@agr.gc.ca (F.M.Y.); sylviej.cloutier@agr.gc.ca (S.C.); 3Crop Development Centre, University of Saskatchewan, Saskatoon, SK S7N 5A8, Canada; valentyna.klymiuk@usask.ca (V.K.); jemanesh.haile@usask.ca (J.K.H.); amidou.ndiaye@usask.ca (A.N.); curtis.pozniak@usask.ca (C.J.P.)

**Keywords:** wheat, genome scans, adaptation, genetic signatures, introgression

## Abstract

Wheat was one of the crops domesticated in the Fertile Crescent region approximately 10,000 years ago. Despite undergoing recent polyploidization, hull-to-free-thresh transition events, and domestication bottlenecks, wheat is now grown in over 130 countries and accounts for a quarter of the world’s cereal production. The main reason for its widespread success is its broad genetic diversity that allows it to thrive in different environments. To trace historical selection and hybridization signatures, genome scans were performed on two datasets: approximately 113K SNPs from 921 predominantly bread wheat accessions and approximately 110K SNPs from about 400 wheat accessions representing all ploidy levels. To identify environmental factors associated with the loci, a genome–environment association (GEA) was also performed. The genome scans on both datasets identified a highly differentiated region on chromosome 4A where accessions in the first dataset were dichotomized into a group (n = 691), comprising nearly all cultivars, wild emmer, and most landraces, and a second group (n = 230), dominated by landraces and spelt accessions. The grouping of cultivars is likely linked to their potential ancestor, bread wheat cv. Norin-10. The 4A region harbored important genes involved in adaptations to environmental conditions. The GEA detected loci associated with latitude and temperature. The genetic signatures detected in this study provide insight into the historical selection and hybridization events in the wheat genome that shaped its current genetic structure and facilitated its success in a wide spectrum of environmental conditions. The genome scans and GEA approaches applied in this study can help in screening the germplasm housed in gene banks for breeding, and for conservation purposes.

## 1. Introduction

Domestication is one of the major bottlenecks in crops that can result in a drastic reduction in genetic diversity compared to the wild gene pool [1]. The loss of adaptive alleles and inbreeding within domesticated plants can amplify the accumulation of deleterious mutations [2,3]. Most grain crops were domesticated mainly for their seed traits, such as seed size, number of seeds per spike or pod, and shattering resistance [1,4,5]. In most cases, they were domesticated as a single or rare event, and might have been limited to only sites of origin in terms of geographic adaptation [1]. However, the preserved resiliencies from the founder seeds, coupled with subsequent post-domestication selections and hybridizations, have enabled the domesticated species to spread to their current broad spectrum of eco-geographic territories far from their origins [6].

Wheat (*Triticum* sp.) is believed to have originated and been domesticated approximately 10,000 years ago in the Fertile Crescent region, alongside other founder crops [7], most likely in the region(s) of present day Turkey and/or Syria [8,9]. Wheat had spread throughout the rest of the Fertile Crescent, in the Near East, Levant, Europe, and Central Asia by about 7000 BC [7,10]. In the next millennium, the crop expanded to the Mediterranean, the Nile-irrigated fields in Egypt, Ethiopia, and India [11] and by around 2000 BC, it had reached as far east as China, likely via Russia [12]. Currently, wheat is produced in more than 130 countries, accounting for more than 25% of total cereal production [13].

The success of wheat, and particularly that of what is commonly known as bread wheat (*T. aestivum*), is incongruent with the short history of its triple bottleneck events: polyploidy, domestication, and transition from the hulled to the free threshing type [14]. Although only a few individuals are assumed to have been involved in interspecific hybridizations between *Aegilops tauschii* (DD genome, 2n = 2x = 14) and *T. turgidum* ssp. *dicoccum*, which resulted in the emergence of hexaploidy wheat [14,15], non-decaying heterosis increased. This increase in non-decaying heterosis resulted in stable vigor and, thence, gene duplication that enabled the individuals to be immune to deleterious mutations [16]. In addition, bread wheat polyploidization occurred by hybridization of genomes adapted to different environments, which contributes to its resilience and adaption to a wide range of environmental conditions [14].

Cultivated wheat exists in three ploidy levels, with seven base chromosomes: diploid (AA genome, 2n = 2x = 14), tetraploid (AABB genome, 2n = 4x = 28) and hexaploid (AABBDD genome, 2n = 6x = 42). These levels are represented by einkorn (*T. monococcum* ssp. *monococcum*), domesticated emmer (*T. turgidum* ssp. *dicoccum*) plus durum (*T. turgidum* ssp. *durum*) and bread (*T. aestivum*) wheat, respectively. Hexaploid wheat accounts for nearly 95% of currently cultivated wheat, while the remaining fraction is mostly durum wheat [17]. The diploid (einkorn) and the domesticated emmer are minor contributors. The genome complexity increases with the ploidy level. The cultivated diploid and tetraploid wheats have close wild relatives: wild einkorn (*T. monococcum* ssp. *boeoticum*) and wild emmer (*T. turgidum* ssp. *dicoccoides*), respectively. The modern tetraploid, durum wheat, was derived from domesticated emmer wheat (*T. turgidum* ssp. *dicoccum*) via selection [18]. However, there is no known close wild relative of hexaploid wheat, which is believed to have arisen, and been domesticated, at the same time.

The post-domestication divergence and success of wheat, in having such a wide eco-geographical range, indicate the evolution of genetic variations into their respective niches. Reconstructing and understanding the past selection and hybridization events of wheat, and tracing adaptive and selection signatures, assist breeding and conservation efforts. Current advances in genome technologies, and in computational and statistical methods in handling big data, have created opportunities to explore historical adaptive signatures and to reconstruct the past selection, hybridization, and other evolutionary events of several species [19,20], including those with complex genomes, such as wheat [18,21,22]. Genome scans, based on FST and principal components, have pinpointed genetic signatures of past selection and hybridization events, as well as the genomic regions underlying adaptation to environmental factors [23,24]. Using whole genome exome capture data, He et al. [22] delimited a large introgression segment on chromosome 4A, between positions 174,725,311 and 485,745,837  bps, and postulated that its origin was wild emmer wheat. Interestingly, this introgression is more frequent in modern cultivars than in landraces; hence, this observation justifies the need for further investigations to trace its origin. The lineage of most modern cultivars can be traced back to the approximately 100-year-old semidwarf cultivar, Norin-10. Norin-10 was developed through a triple crossing of two parents from the US and a Japanese dwarf local variety called Daruma [25]. One of the parents from the US, called Turkey Red, originated from Turkey, one of the havens of wild emmer [26], and was likely moved to the US by Russian settlers in the second half of the 19th century [27]. The Norin-10 dwarf parent, Daruma, should also not be dismissed as a potential source of the introgression, because of its potential lineage with other ancient wheat ecotypes, such as Indian dwarf, Tibetan semi wild, Xinjiang, and Yunnan wheat, in this geographic region [28,29]. The physical position of the introgression, which encompasses the centromeric region, and the historical lineage of modern cultivars derived from Norin-10, suggest its origin from one of the parents of Norin-10, instead of being a direct introgression from wild emmer.

To test this hypothesis, and to gain insights into wheat adaptation in a broader sense, we performed genome scans based on principal components (PCs) and genome–environment association (GEA) analyses using exome capture [22] and whole genome genotyping [21] datasets to do the following: (1) identify adaptive genetic signatures across the wheat genome; (2) reconstruct the genetic divergence throughout its cultivation history; (3) identify genes playing a role in regulating adaptive traits relevant to the resilience of the crop to various environmental conditions. Further analyses were also performed to more specifically address the origin of the previously reported introgression on chromosome 4A.

## 2. Results

### 2.1. Population Structure

Based on a scree plot (Appendix A), the number of appropriate subpopulations (K) for the 921 individuals of the exome capture dataset was estimated to be eight, whereas the landrace subset of the same dataset was estimated to have seven subpopulations (K) (Appendix A). Admixture analysis, based on the corresponding estimated K, showed a similar pattern of clustering in both the main (*n* = 921), (Figure 1a, Appendix A), and the landrace (*n* = 347), (Figure 1b, Appendix A), subsets. Similar clustering patterns were also obtained with the PCA (Figure 1c–f). The NJ phylogenetic tree clustered the accessions into seven major clades in both the main and landrace subsets (Figure 1g,h). Most accessions followed the clustering pattern obtained by the admixture and the PCA analyses. The clusters tended to reflect the geographic origin of the germplasm. Subpopulations dominated by Australian (AUS), Mediterranean (MED) and the former Soviet Union (FSV) accessions were well separated. One of the major Neighbor-joining (NJ) phylogenetic clades exclusively contained nearly 95% Australian germplasm (Figure 1g). There were also other clades where Australian materials were at a higher proportion. The Old-World collection subpopulation (OLDWP) contained germplasm of wide geographic origin, mainly from the Old World, with a long history of wheat production in Asia, Europe and North-East Africa.

The clustering pattern was similar to the overall population, even though most accessions from the New World (e.g., Australia) were not represented in the landrace subset. The accessions in the OLDWP clustered together and occupied a separate clade in the NJ tree (Figure 1h). The MED, East European (EEU), European–Mediterranean (EUMED), the subpopulation with no distinct geographical origin (WLDMix) and FSV were maintained in the landrace subset. A new subpopulation (South Asian; SAS), dominated by materials from South Asia, mainly the Indian subcontinent and its vicinity, was distinct within the landrace subset.

### 2.2. Genetic Variation and Diversity

The largest variation (FST = 0.34) in the overall exome capture population (n = 921) was computed between subpopulations dominated by collections from the FSV and the OLDWP. These two subpopulations also had the first and second highest variations with most other subpopulations (Table 1). The FSV and OLDWP also showed the second highest variation (FST = 0.23) in the landrace subset. These two subpopulations harbored low gene diversity in the overall population. The largest gene diversities for intra- (D_intra_ = 0.50) and inter-individual (D_inter_ = 0.61) comparisons were both observed in the subpopulation where Australian accessions mixed with others (AUSb). In the landrace subset, the largest differentiation (FST = 0.24) was between FSV and the SAS, while the smallest were observed for SAS, followed by OLDWP (Table 2). Despite its small size, the FSV subpopulation had the largest inter-individual gene diversity (D_inter_ = 0.66) in the landrace subset.

### 2.3. Genome Scans and Dichotomy in a Wheat Population

Genome scans on both the main and landrace subsets of the exome capture datasets revealed a highly differentiated genomic region on chromosome 4A spanning an approximately 174–465 Mb region (Figure 2). Both genome scans also identified strongly differentiated loci on chromosomes 2A, 5A, 6A, and 7B, as well as additional noticeably consistent outlier markers on chromosomes 1B, 4B, 5D, 7A and 7D (Figure 2).

The genome scan on the whole-genome dataset magnificently discriminated the readily identified large region on chromosome 4A (Appendix A), which was consistent with the region captured by scans on the exome dataset (Figure 2). The whole-genome data set scan also identified a strongly differentiated region on chromosome 6B (Appendix A). Noticeable outlier regions were also observed on chromosomes 3B, and 4B (Appendix A).

To dissect the genetic variation within the highly differentiated chromosome 4A region, phylogenetic and PC analyses were performed based solely on the SNPs within the outlier region of the exome dataset. Both analyses dichotomized the main and the landrace subset populations into group 1 (G1) and group 2 (G2) (Figure 3c–h). To observe the patterns of variable and fixed evolutionary rates, phylogenetic analyses were performed, based on Neighbor-joining (NJ) and the Unweighted Pair Group Method with Arithmetic Mean (UPGMA), respectively. The NJ tree showed pronounced differences in branch length between the two groups (Figure 3c,d), whereas the UPGMA phylogenetic analysis consistently and clearly dissected the populations into the two groups (Figure 3e,f). This dichotomy could also be observed in the PCAs (Figure 3g,h). The majority of the individuals of the OLDWP subpopulation belonged to G2. A substantial number of Mediterranean accessions were also included in this group. Indeed, G2 included 130–140 of the 347 landraces and 250–260 of the 921 accessions of the overall exome capture dataset.

Haplotype analysis, based on the 62 most extreme outliers (*p* < 1 × 10^−750^) without missing SNPs in the highly differentiated 4A region, divided the genotypes into eight haplotypes where a haplotype was considered if present in at least five individuals. Congruent to the population structure of this region, the haplotypes were also dichotomized in the two previously defined groups, where G1 was dominated by haplotype 1 (Hap1) and G2 by haplotype 2 (Hap2). In G1, 560 individuals had Hap1, while in G2, 115 individuals had Hap2; together, these two main haplotypes represented 675 of the 921 accessions of the overall population (Appendix A). The two haplotypes were well distributed worldwide but a higher frequency of Hap2 was observed in accessions from some Mediterranean and Middle East regions (Figure 4). Nearly all the haplotypes fell under categories of 95% and above similarity in either of these two dominant haplotypes Hap1 and Hap2 (Appendix A).

### 2.4. Genome–Environment Association (GEA)

The GEA of 113 individuals, with geographic coordinates of the exome capture SNP dataset, identified a total of nine loci associated with latitude and temperature records based on the FDR significance threshold (FDR < 0.05). Of these, five were associated with latitude, including three on chromosome 7A, and the remaining two on 6B and 7B (Table 3). The two loci on chromosome 7A, Chr7A:118327401 and Chr7A:141858291, were also significantly associated with latitude based on the Bonferroni threshold (Table 3, Figure 5a), and were close to the moderately differentiated region on this chromosome (Figure 2). The closely-located SNPs, Chr2A:344449220 and Chr2A:372230509, were associated with air temperature, and the former had a *p*-value below the Bonferroni threshold (Table 3, Figure 5b). This locus on chromosome 2A overlapped with the differentiated region identified by genome scans of the overall and landrace datasets (Figure 2). The SNP on 2A, Chr2A:372230509, was located 5 Mb from the strong adaptive signatures of Chr2A:377596631 identified by the genome scan. Other co-located SNPs on chromosome 3B, Chr3B:416694120 and Chr3B:416693883, were associated with earth skin surface temperature (Table 3). However, *p*-values for both SNPs exceeded the Bonferroni-adjusted significance threshold (α/*n* where α = 0.05, *n* = number of makers) (Figure 5c).

### 2.5. Candidate Genes

The significant loci, whether represented by large regions of co-located significant loci or regions surrounding discrete significant loci identified by genome scans, harbored genes potentially important for the adaptation of the crop to different environmental conditions (Table 4). The highly differentiated region on chromosome 4A harbored many genes that could be important for the adaptation of plants to environments. Many of these genes have the potential to be pleiotropic, and, hence, to regulate multiple adaptive traits. This region harbored more than 25 genes playing a role in seed dormancy, more than 30 were flowering time-related genes, and more than 50 genes were associated with abiotic stress responses, mainly to salt, drought, heat and cold (Appendix A). Of these, eight genes were predicted to specifically regulate these three important adaptive traits, namely seed dormancy, flowering time and stress tolerance (Table 4). The region also harbored nine cytochrome genes and 15 genes associated with photosystems (Appendix A).

The 2A:378–574 Mb region harbored *TraesCS2A02G337900* and *TraesCS2A02G339200*, which are predicted to encode AGAMOUS-LIKE NITRATE REGULATED 1 (ANR1) and RIBOSOMAL PROTEIN L34 (RPL34), respectively. The 7B:221_480 Mb locus contained *TraesCS7B02G184800*, predicted to encode an RNA polymerase sigma factor (SIGA). *TraesCS4A02G157100*, located in the 4A region, was also predicted to encode a SIGA, which is an important gene in adaptation to different environmental conditions (Table 4).

The ±200 kb window surrounding the adaptive signature Chr5A:502614628 locus (Figure 2, Table 4) contained *TraesCS5A02G292700* and *TraesCS5A02G292900*, which predictively encode ciliary BASAL-BODY PROTEINS WITH UPREGULATED GENES 22 (BUG22) and CYTOKININ-RESPONSIVE GROWTH REGULATOR (CKG) proteins, respectively (Table 4). A total of six genes, all orthologous to AT1G28670, which encodes *Arabidopsis thaliana* lipase (ARAB-1), were found within the vicinity of the 7A:690367177 locus (Table 4).

The loci captured by the GEA also harbored genes that play potential roles in responding to environmental factors. The Chr2A:344449220-372230509 locus, associated with mean annual temperature, on chromosome 2A, harbored *TraesCS2A02G243800* and *TraesCS2A02G248700*, which predictably encode the heat shock Hsp70-Hsp90 organizing protein and a protein of the CST complex involved in telomere maintenance, respectively (Table 5). The locus marked by the co-localizing loci Chr3B:416693883 and Chr3B:416693839, on chromosome 3B, associated with earth skin temperature, contained *TraesCS3B02G258800* (Table 5), a pleiotropic gene presumed to regulate responses to abiotic stresses (temperature and drought), as well as agronomic and physiological traits, such as grain size, seed dormancy, and phenological traits (e.g., heading date). The other loci captured by the GEA, and associated with latitude, included genes with important roles in plant responses to light and temperature, influencing vital physiological processes, including seed dormancy and flowering-related behaviors. For example, the latitude-associated Chr7A:118327401 locus included *TraesCS7A02G162400* and *TraesCS7A02G162500*, which were orthologous to *AT1G48270* and *AT2G41850*, respectively. These genes are predicted to encode G-PROTEIN-COUPLED RECEPTOR 1 (GCR1) and a pectinlyase-like superfamily protein (ADPG2) involved in floral physiology and seed dormancy, respectively (Table 5).

### 2.6. Potential Source of ‘Introgression’ to the 4A Region

To trace the potential source of DNA segments and possible gene flow signatures at the highly differentiated 4A region, the 5000 SNPs with the most significant *p*-values from the overall exome capture dataset were selected from the significant region on chromosome 4A for F- (FST) and Phi- (PhiPT) analyses. Both FST and PhiPT analyses showed significant (*p* < 0.001) differentiation between wild emmer and hexaploid wheat landraces. The hexaploid cultivars and nearly all the durum accessions were undifferentiated from the wild emmer accessions. In fact, these latter two had the lowest differentiation M (FST = PhiPT = 0) with wild emmer, based on both analyses (Table 6, Appendix A). In contrast, the G-test, after 10,000 Markov Chain and 5000 iterations, indicated strong differentiation between hexaploid cultivars and wild emmer, while the hexaploidy wheat cultivars remained undifferentiated from the durum wheat accessions (Appendix A). Others, such as Indian dwarf (*T. aestivum* L. ssp. *sphaerococcum*), Tibetan semi-wild (*T. aestivum* ssp. *tibetanum*), Georgian wheat (*T. turgidum* ssp. *paleocolchicum/T. karamyschevii*), Xinjiang wheat ((*T. aestivum* ssp. *petropavlovskyi/T. petropavlovskyi*), Persian wheat (*T. turgidum* ssp. *carthlicum*) and Vavilovii wheat (*T. vavilovii*), had similarly low FST and PhiPT (FST = PhiPT = 0) with hexaploid cultivars (Table 6, Appendix A). These wheats were also undifferentiated from the hexaploid cultivars in the G-test (Appendix A).

Based on mean square allele size difference (MSD) of the most significant 5000 SNPs of the 4A region, it appeared that the Xinjiang wheat lacked both intra- and inter-accession gene diversity. Indian dwarf and Vavilovii wheats had the next lowest gene diversity, with MSD of 4.10 × 10^−5^ and 4.12 × 10^−5^, respectively, at the individual and the population levels (Table 6). Durum wheat also showed a comparably low intra-accession gene diversity (MSD = 4.73 × 10^−5^). The highest intra-accession gene diversity was computed for spelt wheat (*T. aestivum* ssp. *spelta*; MSD = ~0.14), followed by macha wheat (*T. aestivum* spp. *macha*; MSD = ~0.12). The highest inter-accession diversity was computed for hexaploid wheat landraces (MSD = ~0.9), followed by Yunan wheat (*T. aestivum* ssp. *yunnannense*; MSD = ~0.68).

After assigning the accessions to five subpopulations (*K* = 5; Figure 6a), ancestral proportion analysis of the 4A region showed that, with the exception of the Canadian cultivar Taes_Cul_CAN-C1, all other wheat cultivars shared more than 95% of their ancestry with wild emmer (Figure 6b, Appendix A). This group of, predominantly, cultivars also included around 58 % of the hexaploid wheat landraces with more than 90% shared ancestral proportion (Appendix A). The remaining landraces appeared in a separate group that had in excess of 90% shared ancestral proportion and that included all spelt and macha wheats (Figure 6b, Appendix A). This group showed an uniquely long branch in the phylogenetic tree (Figure 6c) and its clade was located between the diploid *T. urartu* and *T. monococcum* clades and the clade that grouped the rest of the wheat types (Figure 6c,d). Most members of this group were concentrated in the Mediterranean region of Eurasia, where *T. urartu* and *T. monococcum* can also be found (Figure 6e).

## 3. Discussion

Wheat has successfully spread to a wide range of eco-geographic regions. It is one of the most cultivated crops, and contributes the major proportion of global nutrition, accounting for 20% of the calories and protein of the world’s population. The success of wheat in its wide environmental spectrum is largely attributed to historical selection and hybridization events that shaped its genome into its current genetic structure and variation. Despite hosting the triple genetic bottlenecks of polyploidization, domestication and transition from hulled to free threshing [14], multiple factors, including conditions associated with the bottleneck events, contributed to its success. Polyploidization, and, in particular the case of hexaploids, contribute to gene duplications buffering individuals against deleterious mutations [16]. The bread wheat’s polyploidization, via hybridization of genomes adapted to different environments, increases resilience and improves its adaptability to diverse environmental conditions [14]. Post-domestication human interventions, including selections, hybridizations, and the spread of germplasm, hastened the success of crop plants, in general, and of wheat, in particular, and broadened their range of environments and cultural realms [1]. Continuing post-domestication gene flow from wild relatives also contributes to diversification and adaptation to new niches [30,31].

### 3.1. Population Genetic Structure and Variation

The genetic structure of both populations (overall *n* = 921 and landrace subset *n* = 347) in this study did not perfectly follow the geographic origin of the germplasm, which is inconsistent with previous studies that reported strong geographic differentiation among wheat populations [32,33,34]. Population genetic structures with clusters containing accessions of different geographic origins may reflect historical human-mediated gene flows, attributed to past explorers and breeding-affiliated institutions’ germplasm collections, to trade, and to human migrations [35,36]. The clustering of Australian origin materials within a distinct subgroup, despite the short history of wheat in the region [34], suggests accelerated separation of this germplasm from the founder seeds, which emanated from the major geographic origins of Europe, the Mediterranean and Africa [35]. A separate clade of nearly exclusive Australian origin germplasm, in one of the NJ clades (Figure 1g), can be attributed to improved selections and hybridizations towards materials adapted to Australian environments [37]. This has likely led to the divergence of current Australian accessions from materials only introduced into the region less than two and half centuries ago [33].

The distribution of Former Soviet Union accessions (Figure 1) is consistent with a previous study that showed their association with Eastern European and Middle Eastern collections [38]. The representation of the Former Soviet Union collections (Figure 1g,h) in most of the clusters can be related to the oldest and largest gene bank, i.e., the NI. The Vavilov All-Russian Institute of Plant Genetic Resources (VIR) has been collecting and dispatching materials in the region, and worldwide, throughout the history of crop plant distribution [39]. However, the consistently deviant position of FSV in this study suggests a limited spread of this subpopulation, which might be confined to a particular geographic region. On the contrary, the presence of Mediterranean accessions across most clusters plausibly reflects the historical distribution of the wheat germplasm from this region, based on its importance as a center of origin and diversification of the crop.

The OLDWP subpopulation, representing wide geographic origin, mainly from old wheat producing regions, agrees with wheat’s early trade history, migration, and expansions or empires [11]. The low inter- and intra-gene diversity in this subpopulation can be associated with limited breeding interventions to enhance genetic diversity, which characterizes most recent wheat germplasm, and to limited geneflows [34]. This subpopulation spreads from Ethiopia in the south, via the Middle East, to Uzbekistan and Russia in the north and as far east as China. While the subpopulation is well-membered by the Middle East accessions, e.g., accessions from Iraq and Iran, it lacks individuals from the Mediterranean region, with the exception of a few individuals from Tunisia (Figure 1g,h, Appendix A). This strengthens our argument above that this population was less subject to contemporary breeding involving materials from the Mediterranean and the rest of Europe, and had little contribution from Asian landraces [34,40]. The subpopulation might be uniquely adapted to ecological niches that are common in the countries it spans.

Despite the short history of wheat in Australia, its highly diverse germplasm is likely attributable to multiple introductions, hybridizations and introgressions from contemporary breeding activities [35,41]. Wheat was one of the earliest crops in modern breeding [42]. The early wheat breeding programs used germplasm of broad geographic origins, and, not surprisingly, the cultivars released at that time harbored high genetic diversity [35]. The Australian wheat germplasm is also consistent with the diversity of other crop germplasm, such as barley, on the continent [43].

### 3.2. The Differentiation on Chromosome 4A and Adaptive Signatures

Studies have already reported the uniqueness of chromosome 4A, which is attributed to the translocations it received from chromosomes 5A and 7B, and further rearrangements that it underwent [44,45]. Such ancient translocations might have been reflected in all wheat accessions and seldom affect the genetic variation within the species. Recently, the signature on chromosome 4A has been proposed to be an introgression from wild emmer [22]. This region has likely undergone post-domestication selection pressure and a hybridization target and led genetic divergence among current wheat accessions. Hence, the differentiation among the current accessions within the chromosome 4A signature region likely represents recent variations. One may hypothesize that this region is associated with the adaptation of the crop to a wide eco-geographic range. The clear separation into two unique subpopulations of both the overall (*n* = 921) and the landrace subset (*n* = 347) populations in this region of chromosome 4A indicate a major event on the chromosome and also in wheat genome evolution. More interestingly, the dichotomy of both population sets into a group dominated by OLDWP and one containing all remaining accessions invites further investigation of this separation event.

### 3.3. Candidate Genes at Potential Adaptive Loci

The chromosome 4A signature region has a high gene density and contains many genes with potential to affect major target traits of adaptations to environmental conditions, and, as such, may have been subjected to high selection pressure. He et al. [22] previously identified a major ATP-binding cassette (ABC) transporter gene in this region. In the present study, *TraesCS4A02G157100* in the 4A and *TraesCS7B02G184800* in the 7B signature regions were both predicted to encode an RNA polymerase sigma factor (SIGA). The SIGA factor regulates the circadian clock in wheat [46], a vital physiological process for adaptation to environmental conditions [47,48,49]. Individual SIGAs work independently, or in tandem with other SIGAs, to regulate multiple biological processes [50,51]. The *RETINOBLASTOMA*-*RELATED* (*RBR1*) gene in this region may also play a role in regulating biological processes, including DNA repair and hypersensitivity associated with the adaptation of plants to different environmental stresses, such as salinity, temperature, and heavy metals [52]. The RBR1 gene is involved in cell proliferation and regulating responses to DNA damage during biotic and abiotic stresses [53,54].

The 4A signature region also harbored multiple *CALCINEURIN B–LIKE* (*CBLs*) genes interacting with *PROTEIN KINASE* (*CIPK*). The *CIPKs* play pivotal roles in ion homeostasis, in responses to diverse abiotic stresses, including drought and extreme temperature conditions, and in regulating several biological processes that involve the calcium ion [55,56]. The differential CBL regulation in stress conditions [56,57] can lead to adaptive divergence among populations exposed to different environmental niches with varied moisture and temperature ranges [58,59,60].

The *AGAMOUS LIKE 30* (*AGL30*) at this locus also has an important role in pollen development [61] and in the regulation of responses to abiotic stresses [62,63]. The AGLs are essential in regulating pollen viability and pollen tube growth [64,65], including those in wheat [66], and in floral organ development [67]. The AGLs can also regulate seed maturation and activation of dormancy [68]. The predicted brassinosteroid receptor kinase (BRI1) gene in this 4A region also regulates seed germination and dormancy release [69]. Furthermore, BRIs play a role in mediating responses to diverse abiotic stresses, including drought, salt, temperature, and heavy metals [70,71,72,73]. The *TraesCS4A02G182900*, which is predicted to encode the SALT-INDUCED ZINC FINGER PROTEIN2 (SIZ2) has been shown to play a role in tolerance to drought [74] and heavy metals [75], as well as being involved in regulating phenological traits, such as flowering [76]. The involvement of these genes (*CIPK*, *AGLs*, *BRIs*, and *SIZs*) in many critical biological processes renders them not only candidates for natural selection, but also candidates with potential to influence important agronomic traits, such as yield. As such, they could have been the object of anthropogenic (improvement) selections.

The *TraesCS5A02G292900* at Chr5A, 502614628, whose Arabidopsis ortholog is *AT5G50915*, is predicted to function as a *CYTOKININ*-*RESPONSIVE GROWTH REGULATOR* (*CKG*). This gene can be a key player in the adaptation of plants, including wheat, to both biotic and abiotic stresses [77] via biosynthesis of adaptive compounds and signal transduction processes [78,79]. The second candidate gene at this locus, *TraesCS5A02G292700* (Arabidopsis ortholog *AT3G12300*), encodes ciliary protein BUG22 (a basal-body protein with upregulated genes), which is involved in the biogenesis of motile organelles, such as centriole cilia and flagella [80]. Although terrestrial higher plants lack these organelles, ciliary proteins, such as BUG22 [81,82,83], were preserved. These genes are highly expressed in higher plant matured pollen [83], and they may mediate functions in pollen germination and pollen tube growth [84], which are critical processes for the success of plants, including cereals [85], in extreme environmental conditions [86,87,88]. Although these genes encode highly conserved proteins, differences in expression levels among accessions has been observed [81].

The presence of the predicted cytochrome P450 gene *TraesCS6A02G286600LC* at 6A:233244464 hints at the importance of this locus in adaptation to diverse stress conditions [89,90] and morphological regulations [91]. This prediction, however, is one of low confidence. Hence, it is important to obtain additional details for better prediction.

The loci associated with latitude can be attributed to differential factors, such as light photoperiod and temperature. For example, *TraesCS7A02G162400* (Arabidopsis ortholog *AT1G48270*) at 7A: 118327401, is predicted to encode a guanine nucleotide-binding protein G-PROTEIN-COUPLED RECEPTOR 1 (GCR1), which regulates flowering time and seed dormancy, these being typical adaptive responses to environmental conditions, such as seed dormancy break and early flowering [92]. The GCRs are involved in several regulatory and signaling biological processes in response to environmental stimuli, such as light [93], including ultra-violet rays [94] and biotic stress via the cell wall [93,95]. The GCRs operate with the G-PROTEIN Gα (GPA1) to regulate the expression of many genes [93,96]. The co-located *TraesCS7A02G162500*, which is orthologous to *AT2G41850*, is predicted to encode a POLYGALACTURONASE (PG) that is involved in floral and foliar abscissions [97]. Organ abscission, including flowers and leaves, is a mechanism of adaption to environmental conditions because it constitutes a signal to complete the life cycle, often through seed development that perpetuates the germline [98,99,100].

The genes, *TraesCS2A02G243800* and *TraesCS2A02G248700*, associated with temperature, at Chr2A:344449220 and Chr2A:372230509, respectively, are predicted to encode proteins that mediate responses to heat. While the former is predicted to encode Hsp70-Hsp90 organizing protein 1, that regulates heat shock protein chaperones Hsp70 and Hsp90 [101], the latter is orthologous to *AT1G56260* (*MDO1*, *MERISTEM DISORGANIZATION 1*, *TELOMERIC PATHWAYS IN ASSOCIATION WITH STN1*, *TEN1*), a gene that protects against heat-induced telomere truncation and protects against other damage caused by heat [102].

### 3.4. Gene Flow and Source of ‘Introgression’

The observed low genetic differentiation of the hexaploid cultivars and the durum wheats with the wild emmer wheats in the 4A divergent region is consistent with He et al. [22], who interpreted this observation as indication of an introgression from wild emmer into cultivated bread wheat. However, the high proportion of shared ancestry and the low differentiation between hexaploid cultivars and durum wheats with other wheat types, such as Indian dwarf, Tibetan semi-wild, Georgian emmer wheat, Xinjiang wheat, Persian wheat, and Vavilovii wheat, suggest another intermediate source of the 4A introgression, instead of a direct gene flow from wild emmer. The G-test clearly discriminated the cultivars from the wild emmer group (Appendix A), while the above-listed wheat types remained undifferentiated with cultivars. Therefore, this introgression likely arose in modern cultivars from another cultivated wheat type rather than wild emmer. This ‘introgression’ might be associated with the modern breeding tendency towards semidwarf cultivars that came to the attention of breeders with the release of the Japanese cultivar, Norin-10, about a century ago [25]. Since the release of cv. Norin-10, materials carrying its semidwarf genes have spread across the world through germplasm exchanges and were incorporated in modern cultivars, particularly after the green revolution [103]. These historical fingerprints suggest that the source of introgression of this 4A signature region was more likely one of the three parental lines of Norin-10 than wild emmer. The likely maternal line and the dwarf Japanese native variety Daruma might be the source of the genome segment of the 4A high peak region for modern cultivars, considering the position of this region towards the centromere where introgression via recombination is less likely to occur.

The deviance of many wheat landraces from the cultivars, emmer and most other wheat types, coupled with the close genetic relationship of these landraces with spelt and macha wheats, triggers the question as to the chronology of the events that led to the dichotomized separation of wheats in this 4A region. This signature region may indicate the history of spelt wheat and its potential contribution in the bread wheat hybridizations [15,104,105]. Given the fact that the A genome is ancestrally linked to wild emmer, and the fact that the peri-centromeric position of the 4A signature region would be less influenced by subsequent recombination, the segment from wild emmer might be descended from the time of domestication and preserved in cultivars and most other wheats. On the other hand, an introgression and/or other chromosomal change, such as translocation, could have occurred in spelt wheat, which might also be the source of the genome of landraces which are undifferentiated with spelt in the 4A signature region. The geographic distribution of the accessions carrying the aberrant segment is also consistent with variation between European and Asian spelts [105]. The long branches of the clade occupied by the landrace–spelt group can reflect the number of nucleotide substitutions and, hence, the evolutionary age [106] and/or the number of changes [107,108].

Aberrantly differentiated genomic regions are a frequent occurrence in highly-bred- crops, including cereals. These regions usually harbor major adaptive genes and are subjected to both improvement and natural selections. For instance, the SUB1 locus in rice is known for its high level differentiation and contains a gene conferring submergence tolerance [109]. Bekele et al. [110] identified multiple conspicuously differentiated regions in oats. The two most differentiated regions were the *Mrg28* locus, which is an improvement selection target for heading date [111], and the *Mrg18* locus, that harbors multiple disease resistant genes [112,113].

## 4. Materials and Methods

### 4.1. Datasets

In this study we used two SNP datasets. First, approximately 8.7M SNPs, obtained from exome capture data of 1026 wheat accessions comprising landraces, breeding materials, and cultivars, were downloaded, along with the passport data of the accessions, from https://triticeaetoolbox.org/wheat (accessed on 11 August 2022) [22]. Second, approximately 100M SNPs, obtained from whole genome resequencing of cultivated *Triticum* species and other non-domesticated *Triticum* from a primary gene pool, such as *T. monococcum* or *T. turgidum* ssp *dicoccoides*, were downloaded from Genome Variation Map (https://bidg.big.ac.cn/gvm, accessed on 29 August 2022) [21]. We focused on the exome capture data for most analyses because it was derived principally from hexaploid (*T. aestivum*) wheat. The two datasets were processed separately. For both datasets, SNPs on each chromosome were filtered using a 90% call rate. The filtered SNPs from all chromosomes were merged into a single variant call file (VCF) and further filtered to a maximum of 5% missing calls. Individuals missing more than 10% of the SNPs after consolidation of the SNPs from all chromosomes were omitted. The filtered quality-controlled exome capture-derived dataset contained around 113K SNPs and 921 individuals. The whole genome resequencing dataset represented different species, all three ploidy levels (diploid, tetraploid and hexaploid) and genomes (A, B, D). As such, different SNP subsets were generated for downstream analyses. These are described below.

Using the information of the passport data, 347 landraces, selected from the 921 accessions of the exome dataset, were used for downstream analyses to shed light on the historical selection events and eco-geographic adaptation signatures across the habitats of the crop. The final 921 accessions were collected from 96 countries, of which 80 were also the origin of at least one landrace. The germplasms’ countries of origin were positioned on the world map. along with the corresponding annual temperature, using the Quantum Geographic Information System (QGIS) from the Open-Source Geospatial Foundation Project QGIS 3.28 (http://qgis.osgeo.org, accessed on 1 September 2022; Appendix A). To obtain further insights into the adaptation of the germplasm to diverse environmental conditions, 113 landraces with geographic coordinates were selected. Using the geographic coordinates, climate and soil data were downloaded from the NASA website (https://power.larc.nasa.gov, accessed on 22 September 2022). These data, and latitudinal information provided within the passport data of the accessions of this collection [21], were used for genome–environment association (GEA) analyses.

### 4.2. Population Structure and Differentiation Analysis

To define the appropriate number of ancestral subpopulations, the eigenvalues of each individual were estimated for the first 30 PCs and visualized in scree plots using PCADAPT in R [114], separately, for main (*n* = 921) and landrace subset (*n* = 347) populations of the exome capture dataset. Following the Cattle rule [115], as suggested by Luu et al. [114], the PC to the left of the last curve on the scree plot was defined as having the appropriate number of subpopulations (K) for each of the population sets. To cluster accessions based on their genetic relationships, ancestral coefficients (Q) were computed, based on the corresponding estimated number of subpopulations (K) for the main (*n* = 921) and its landrace subset (*n* = 347) using ADMIXTURE v1.3 with default parameters [116]. Subpopulations were assigned to the geographic region from where the majority of the members were collected. Individuals sharing more than 50% of the ancestral coefficient (*Q* ≥ 0.5) at the selected K were assigned to the same group (subpopulation). Individuals with <50% ancestral coefficient were also assigned to the subpopulation where they had the highest ancestral proportion. Following the assignment of all individuals to their appropriate group, the ancestral coefficients were summarized into a structure plot using Tess3 in R [117].

Principal component analysis (PCA) was also performed by setting the number of PCs equal to the number of the subpopulations (K). The first three PCs, along with the percent of variation that each explained, were visualized in a scatter plot using ggplot2 in R. To gain insights into the pattern of clustering without predefining the subpopulations and to hint at the evolutionary events over time, a neighbor-joining (NJ) phylogenetic analysis was performed on the entire population sets from the exome dataset. TASSEL 5.2 [118] was employed for all phylogenetic analyses and the outputs were, subsequently, processed and visualized using interactive Tree of Life (iTOL) [119]. To compute the differentiation among the subpopulations, the program GENEPOP v4.7.5 [120] implemented in R [121] was used. To shed light on the historical bottlenecks and inbreeding events, individual (intra-accession) and subpopulation-wide (inter-accession) gene diversities were calculated.

### 4.3. Genome Scanning for Adaptation Signature and Linked Candidate Genes

To trace genomic regions that had undergone selection and/or that indicated adaptive signatures, a PC-based genome scan was performed using PCADAPT v4 implemented in R [114,122] for the two exome capture population sets, the main (*n* = 921) and the landrace (*n* = 347) subsets, where the number of PCs was equal to the number of subpopulations. To further trace potential adaptive alleles to specific environmental conditions, individuals with known geographic coordinates of the collection site were selected and a GEA analysis was performed using the latent factor mixed model (LFMM2) [123], based on climate and soil data downloaded from the NASA database (https://power.larc.nasa.gov, accessed on 22 September 2022) described above. For genome scans from PCADAPT, genomic regions with strongly differentiated outlier loci were considered without defining a threshold. For GEA, the significance threshold was set to the Benjamini–Hochberg false discovery rate (FDR) [124] at α = FDR < 0.05. Only the first 5 markers in the ascending order of the *p*-values with FDR < 0.05 threshold were considered to be significantly associated. To visualize the overall results, −log10 *p*-values were displayed in Manhattan plots with a threshold mark based on the Bonferroni multiple test correction at α/*n*, where α < 0.05 and *n* = total number of SNPs used for the association analysis.

### 4.4. Gene and Functional Annotation Analysis

To identify potential genes linked to the differentiated regions and loci identified by PCADAPT and GEA, all genes within 200 kb upstream and downstream of the detected loci, and their predicted functional annotations, were extracted from the WheatGmap database (https://www.wheatgmap.org, accessed on 4 October 2022). Where a big chunk of genome region peaked up, all genes within that region were downloaded. The extracted genes’ physical positions matched the coordinates in all wheat genes downloaded from the joint Genome Institute database (https://phytozome-next.jgi.doe.gov, accessed on 9 October 2021). The genes’ functions and the traits affected were further explored using the published literature and a variety of databases, such as ensemblplants (https://plants.ensembl.org, accessed on 21 February 2023) and knetminer *T. aestivum* (https://knetminer.com/Triticum_aestivum, accessed on 21 February 2023). The number of consistent reports and the likelihood of association of a gene with an important adaptive trait was evaluated, based on its predicted function, the strength of the evidence and on the number of reports demonstrating its causal effect.

### 4.5. Source of ‘Introgression’ and/or Genome Segments of Mass Peak Regions

The genome scans from both population sets of the exome capture dataset identified the uniquely differentiated region on chromosome 4A (see Section 2, which had already been assumed to be introgression by He et al. [20]. In search of the potential origin of introgression in this region and to identify further outlier regions, the whole-genome genotyping dataset of different wheat types was utilized. A PC-based scan was performed and regions with a chunk of high-peak markers were selected. Depending on the size of the region, the 3–5K markers with the lowest *p*-value from the genome scan analysis were used to compute FST, PhiPT, and individual ancestral coefficients for the whole genome dataset comprising all ploidy levels and all available wheat types, including cultivars, landraces, other ecotypes and wild relatives. To overcome the sample size differences, population differentiation analyses, including FST and PhiPT, were performed with 10,000 permutations. In addition, a population differentiation analysis based on a maximum likelihood statistical significance G-test was performed after 10,000 Markov chains and 5000 iterations using R package genepop [121].

## 5. Conclusions

The public availability of data in repositories, combined with statistical models and computing abilities, provide opportunities for scientists to pose new hypotheses, deepen analysis and explore new research avenues to address evolution, breeding and conservation in plants and animals. In the present study, genome scan and genome–environment association identified new and previously reported genomic regions in wheat genomes indicative of historical hybridization and evolutionary events and adaptation signatures of the crop across its habitats. Further analyses to confirm the source of introgressions previously postulated to be of wild relative origins is warranted because our results suggest that cultivated wheat ecotypes are a more likely source. Investigations of other genomic regions previously reported as introgressions from the wild are of interest to determine the extent of such potential phenomena and their impacts on wheat adaptation. These considerations within the pedigrees of current cultivars would enable us to revise their genetic potential towards the development of better adapted varieties.

## Figures and Tables

**Figure 1 ijms-24-08390-f001:**
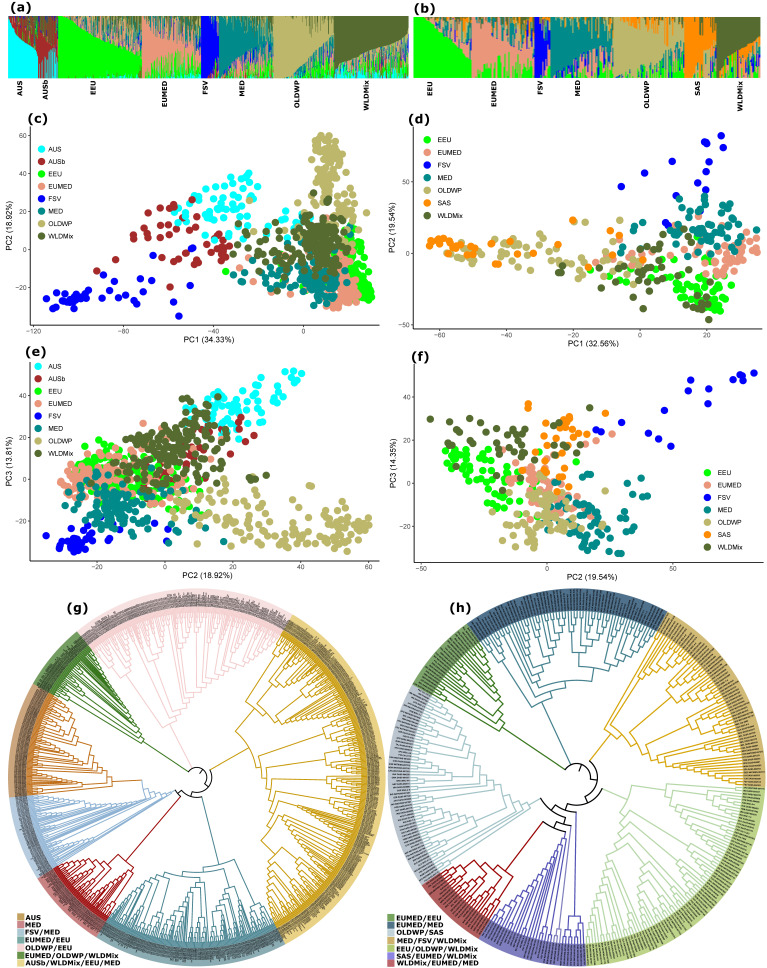
Population structure of the exome capture datasets containing all 921 accessions (**a**,**c**,**e**,**g**) and the 347 landrace subset (**b**,**d**,**f**,**h**). Admixture graphs (**a**,**b**), principal component analysis with the first two principal components (PCs) (**c**,**d**) and with the second and third PCs (**e**,**f**), and neighbor-joining phylogenetic trees (**g**,**h**), are displayed to illustrate the population structure analyses. The proportion of the variation explained by the PCs is indicated on the axes (**c**–**f**). The individual names at the end of the branches are in accordance with the following nomenclature: the first three letters indicate the country of origin, based on UN country coding with some exceptions (e.g., FSV = Former Soviet Union), followed by the maintainer and the unique accession identifier. Therefore, AUS = Australian, AUSb = Australian mixed with others, EEU = East European, EUMED = Europe-Mediterranean mixed, FSV = Former Soviet Union, MED = Mediterranean, OLDWP = from historical wheat producing OLD-WP regions, SAS = South Asian, and WLDMix = with no distinct geographic region.

**Figure 2 ijms-24-08390-f002:**
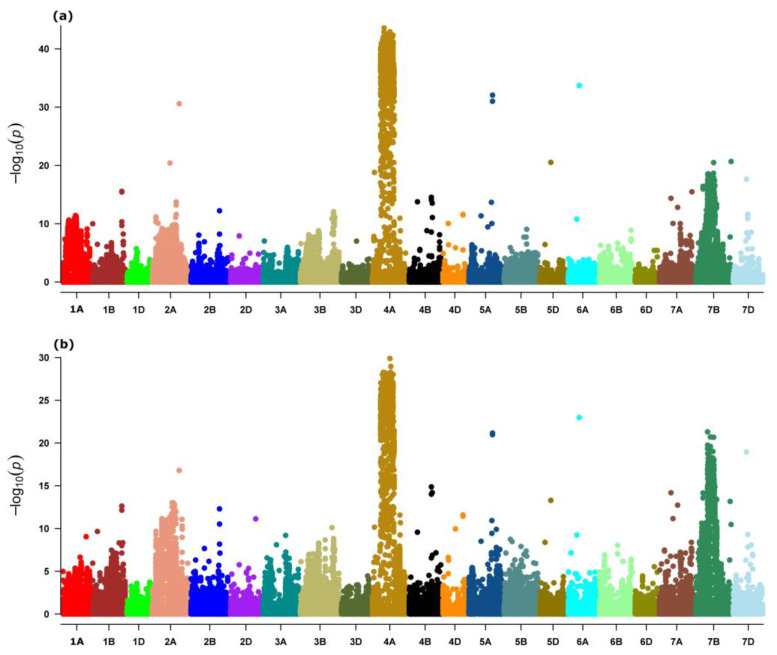
Manhattan plots showing the significant outliers and outlier regions of the wheat genome using a genome scan of (**a**) overall exome capture population (*n* = 921) and (**b**) its landrace subset (*n* = 347).

**Figure 3 ijms-24-08390-f003:**
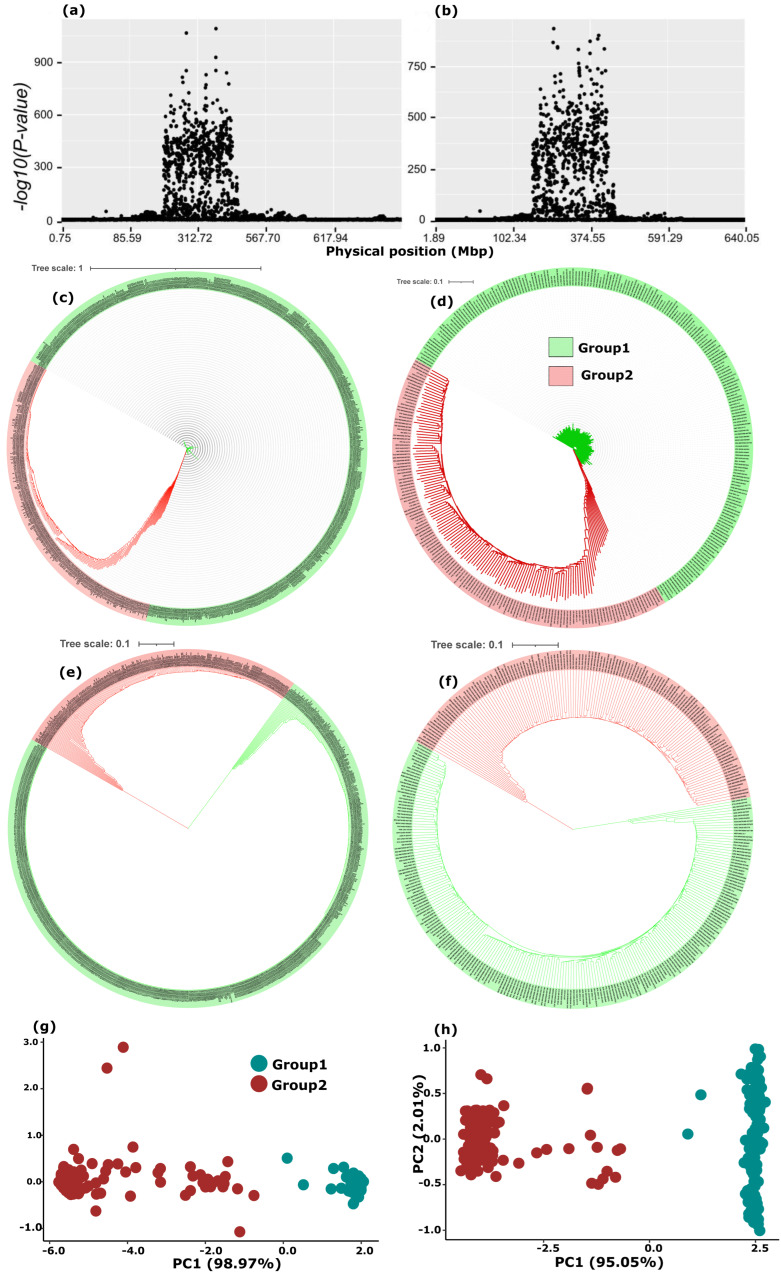
Population structure based solely on the 62 most significant outlier single nucleotide polymorphism markers (*p* < 1 × 10^−750^) located in the differentiated region of chromosome 4A. All figures on the left (**a**,**c**,**e**,**g**) are for the main subset (*n* = 921) and on the right (**b**,**d**,**f**,**h**) for the land race subset (*n* = 347). (**a**,**b**) show the differentiated region on chromosome 4A, (**c**,**d**) show the neighbor joining phylogenetic tree showing the branch length, while (**e**,**f**) show the topology of the neighbor joining phylogenetic tree and (**g**,**h**) show the principal component analyses based on the first two PCs.

**Figure 4 ijms-24-08390-f004:**
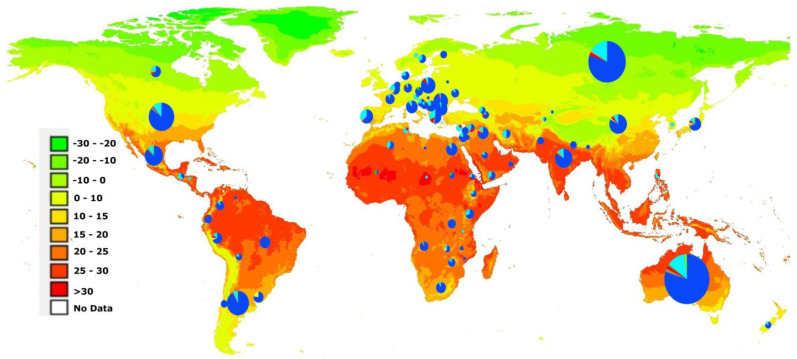
Geographic distribution of the eight haplotypes carried by more than five individuals and defined by their genotypes at the 62 most significant outlier single nucleotide polymorphism markers overlaid on the world map, which are colored according to mean annual temperature (°C). The circle size is proportional to the number of accessions and the pie colors indicate the haplotypes: Hap1 in dark blue and Hap2 light blue.

**Figure 5 ijms-24-08390-f005:**
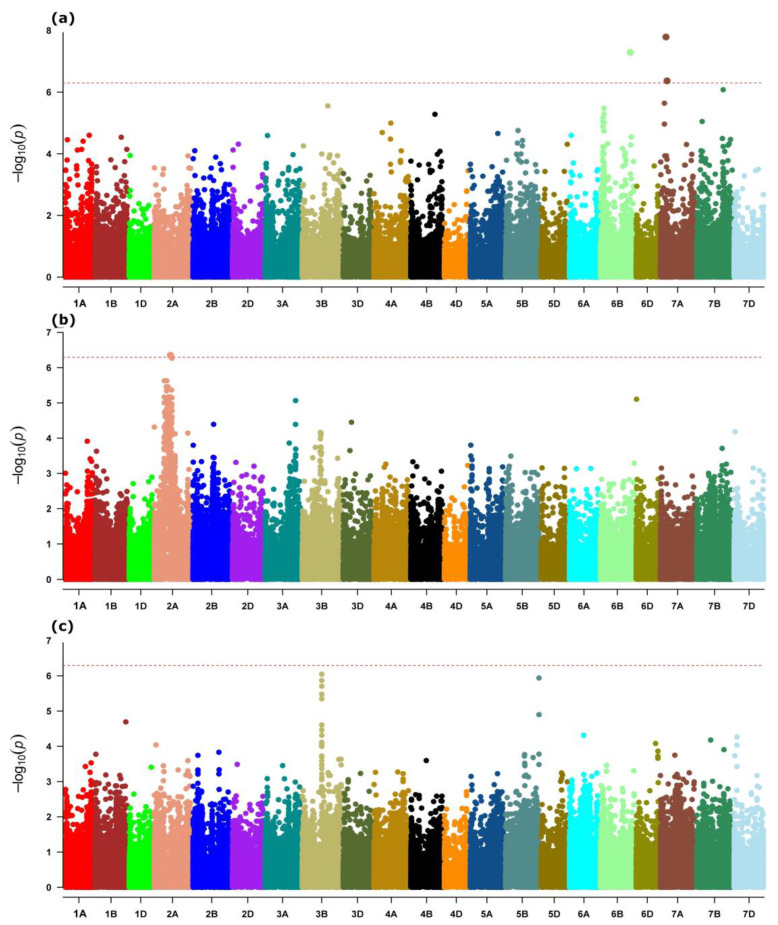
Manhattan plots showing genome–environment association (GEA) for (**a**) latitude, (**b**) mean annual temperature, and (**c**) mean annual earth skin temperature.

**Figure 6 ijms-24-08390-f006:**
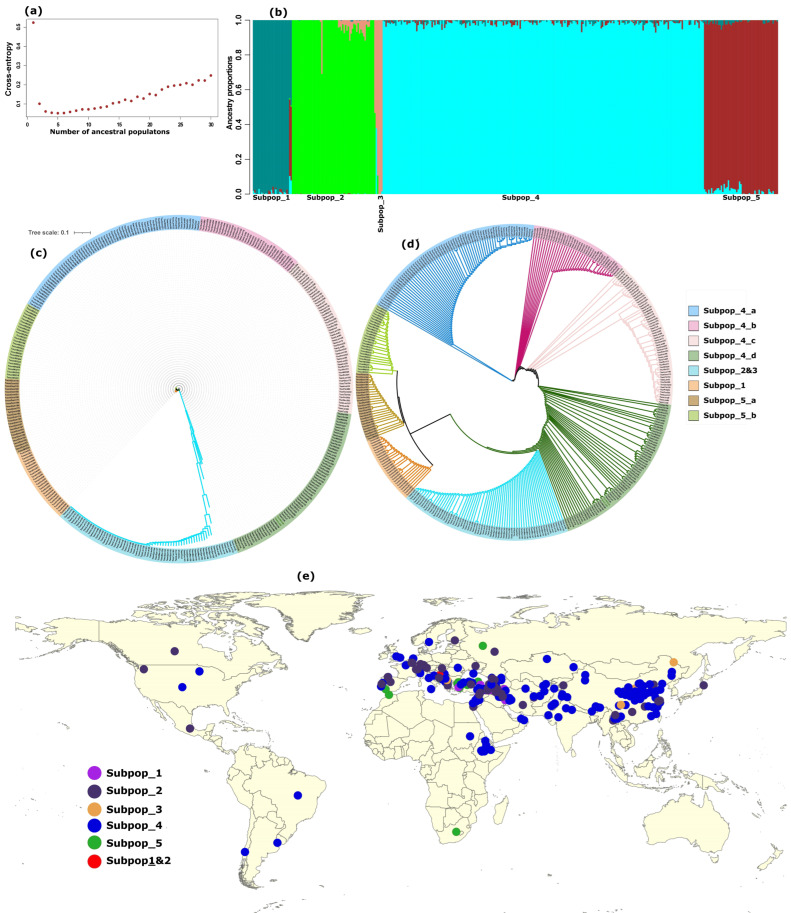
Wheat population structure and geographic distribution of accessions in the whole genome genotype dataset. (**a**) ancestral entropy, shows five ancestral populations (*K* = 5); (**b**) population structure (bar plots), shows percent ancestral coefficient shared among individual members of the populations: Subpop1 = *T. urartu*, Subpop2 = bread wheat landraces (~40%), and other wheat types, such as club, macha and spelt; Subpop3 = bread wheat landraces; Subpop4 = All *T. turgidum* (durum, wild and domesticated emmer) + all bread wheat cultivars + the majority (58%) of the bread wheat landraces + Asian varieties (e.g., Indian dwarf, Chinese ecotypes such as Tibetan semi-wild, Yunan and Xinjiang wheats); Subpop5 = *T. monococcum* (both domesticated and wild einkorn). (**c**,**d**) neighbor joining phylogenetic trees, shows the evolutionary history and topology, respectively and texts at the end of each branch indicate the wheat types. (**e**) the geographic distribution of individuals within each subpopulation, wherein Subpop1 and Subpop2 include individuals with noticeable admixture (>30%) between *T. urartu* and bread wheat landraces.

**Table 1 ijms-24-08390-t001:** Pairwise genetic differentiation (FST) and inter- and intra-individual gene diversity in the overall exome capture population (*n* = 921).

POP	AUS	AUSb	EEU	EUMED	FSV	MED	OLDWP	WLDMix
AUS	0.00							
AUSb	0.08	0.00						
EEU	0.16	0.18	0.00					
EUMED	0.15	0.16	0.04	0.00				
FSV	0.28	0.17	0.33	0.30	0.00			
MED	0.14	0.14	0.06	0.04	0.27	0.00		
OLDWP	0.16	0.18	0.10	0.10	0.34	0.10	0.00	
WLDMix	0.10	0.12	0.05	0.04	0.29	0.06	0.08	0.00
D_intra_	0.44	0.50	0.44	0.43	0.38	0.44	0.35	0.44
D_inter_	0.54	0.61	0.53	0.56	0.54	0.58	0.51	0.58

D_intra_ = intra-individual genetic diversity, D_inter_ = inter-individual gene diversity; subpopulations: AUS = Australian, AUSb = Australian mixed with others, EEU = East European, EUMED = Europe-Mediterranean mixed, FSV = Former Soviet Union, MED = Mediterranean, OLDWP = from historical wheat producing OLDWP regions and WLDMix = with no distinct geographic region.

**Table 2 ijms-24-08390-t002:** Pairwise genetic differentiation (FST) and inter- and intra-individual gene diversity in the exome capture landrace subset (*n* = 347).

POP	EEU	EUMED	FSV	MED	OLDWP	SAS	WLDMix
EEU	0.00						
EUMED	0.05	0.00					
FSV	0.21	0.17	0.00				
MED	0.08	0.04	0.16	0.00			
OLDWP	0.11	0.11	0.23	0.10	0.00		
SAS	0.16	0.15	0.24	0.15	0.09	0.00	
WLDMix	0.05	0.06	0.19	0.08	0.11	0.13	0.00
D_intra_	0.45	0.43	0.42	0.44	0.37	0.33	0.41
D_inter_	0.53	0.56	0.66	0.57	0.50	0.46	0.54
POP	EEU	EUMED	FSV	MED	OLDWP	SAS	WLDMix

D_intra_ = intra individual genetic diversity, D_inter_ = inter individual gene diversity, Subpopulations: EEU = East European, EUMED = Europe-Mediterranean mixed, FSV = Former Soviet, MED = Mediterranean, OLDWP = subpopulation comprising accessions from historical wheat producing regions, SAS = South Asian (mainly from Indian subcontinent), and WLDMix = Subpopulation with no distinct geographic region.

**Table 3 ijms-24-08390-t003:** Loci associated with environmental factors captured by Genome–Environment Association (GEA), based on the Benjamini–Hochberg significance threshold (FDR < 0.05).

Marker	Chr	Pos	Env	*p*-Value	Order	FDR
6B_637421616 *	6B	637421616	Lat	5.16 × 10^−8^	2	0.003
7A_85621619	7A	85621619	Lat	2.29 × 10^−6^	5	0.045
7A_118327401 *	7A	118327401	Lat	1.63 × 10^−8^	1	0.002
7A_141858291 *	7A	141858291	Lat	4.31 × 10^−7^	3	0.014
7B_559125228	7B	559125228	Lat	8.36 × 10^−7^	4	0.021
2A_344449220 *	2A	344449220	Temp	4.44 × 10^−7^	1	0.044
2A_372230509	2A	372230509	Temp	5.42 × 10^−7^	2	0.027
3B_416693839	3B	416693839	TS	1.97 × 10^−6^	4	0.049
3B_416693883	3B	416693883	TS	1.35 × 10^−6^	3	0.044

* SNPs with *p*-value below the Bonferroni significant threshold (α/*n*, where α = 0.05, *n* = number of makers); Chr = Chromosome; Pos = the physical position of the SNP on the chromosome in base pair; Env = Environmental factor associated with the genetic data, Order = the position of the SNP in the ascending order of all *p*-values of all SNPs under analysis; FDR = false discovery rate based on Benjamini-Hochberg; Lat = latitude; Temp = mean annual air temperature at 2 m above ground, TS = mean annual earth skin temperature.

**Table 4 ijms-24-08390-t004:** Potential adaptive genomic regions captured by genome scans and genes with putative roles in adaptation harbored at these loci.

Locus ^1^	Chr	Gene Accession	Pos ^2^	Gene-Name	Role ^3^
1B:620127043-8142	1B	*TraesCS1B02G386200* *	620129121	*H2B*	
2A:378–574 Mb	2A	*TraesCS2A02G337900*	572054907	*ANR1*	1, 2, 3, Cr
		*TraesCS2A02G339200*	573335887	*RPL34*	1, 2
		*TraesCS4A02G157100*	318034542	*SIGA*	1, 2, 3, Cr
4A:175–500 Mb	4A	*TraesCS4A02G136500*	191118052	*CIPK2*	1, 2, 3
		*TraesCS4A02G143200*	235648062	*RBR1*	1, 2, 3
		*TraesCS4A02G169000*	423891230	*AGL30*	1, 2, 3
		*TraesCS4A02G182900*	460229640	*SIZ2*	1, 2, 3
		*TraesCS4A02G190400*	469100913		1, 2, 3
		*TraesCS4A02G194800*	476977837	*CIPK2*	1, 2, 3
		*TraesCS4A02G200000*	488110425	*CIPK6*	1, 2, 3
		*TraesCS4A02G206200*	499341057	*CPK4*	1, 2, 3
4B:173356497	4B	*TraesCS4B02G133700*	173354146	*4CL1*	
		*TraesCS4B02G133400*	173174111	*UBL5*	
		*TraesCS4B02G133600*	173299661	*SMU2*	
Chr5A:502614628 (502524203-502614764)	5A	*TraesCS5A02G293000*	502798626	*AGD1*	1, Y, PH
		*TraesCS5A02G292900*	502785905	*CKG* (*bHLH*)	3
		*TraesCS5A02G292200* *	502531277	*RGA1*	3, 2
		*TraesCS5A02G292600* *	502597149	*UBQ13*	2
		*TraesCS5A02G292700*	502664303	*CFAP20* (*BUG22*)	
		*TraesCS5A02G292500* *	502584783	*SRK6*	
		*ENSRNA050011189* *	502580696	*TRNA*-*ASN*	
Chr6A:233244464	6A	*TraesCS6A02G286600LC*	233248578	*CP450*	
(233244453-233246282)					
7A:245722423	7A	*TraesCS7A02G256600*	245594974	*GAE6*	4
7A:690367177		*TraesCS7A02G500000*-*6000*	690439204	*ARAB*-*1*(*6*)	1, 2, 3, Y
7B:221–480 Mb	7B	*TraesCS7B02G183800*	292828452	*GRF1*	1, 2, 3, 4
		*TraesCS7B02G184800*	299246343	*SIGA*	1, 2, 3
		*TraesCS7B02G188000*	317228074	*LHY*	1, 2, 3
		*TraesCS7B02G204900*	375199415	*MYBS3*	1, 2, 3
		*TraesCS7B02G211600*	387160478	*BHLH3*	1, 2, 3

^1^ Numbers after the colons indicate the position of the most significant SNPs or the genomic regions (approximate range in megabase) harboring multiple co-located significant loci on chromosomes 2A, 4A and 7B. The numbers in brackets indicate the LD block (r = 0.7) that flanks the loci. ^2^ Starting position of the gene; ^3^ Major traits of potential selection targets 1 = seed dormancy, 2 = flowering date; 3 = abiotic stresses, 4 = disease resistance Cr = Circadian rhythm, Y = yield, PH = plant height. * genes within LD block of the detected loci.

**Table 5 ijms-24-08390-t005:** Potential adaptive genes harbored at loci associated with environmental factors.

SNP ^1^	Chr	Trait	*p*-Value	Gene Accession	Gene Name	Gene Pos ^2^	Potential Target Traits Affected
Chr2A:344449220	2A	Temp	4.44 × 10^−7^	*TraesCS2A02G243800* *	*HOP1*	344447847	Heat responses
Chr2A:372230509	2A	Temp	5.42 × 10^−7^	*TraesCS2A02G248700*	*CST*	372370888	
Chr3B:416693883	3B	TS	1.35 × 10^−6^	*TraesCS3B02G258800* *	*EOBI*	416513594	Seed dormancy, abiotic stress (drought, salt, cold)
				*TraesCS3B02G258900* *		416564891	Proline content (abiotic stress)
Chr3B:416693839	3B	TS	1.97 × 10^−6^	*TraesCS3B02G259000* *	*SNL6*	416690441	Abiotic stress (drought), plant height
				*TraesCS3B02G259100*	*PERK11*	416805836	Male sterility, root hair length
Chr6B: 637421616	6B	Lat	3.37 × 10^−6^	*TRAESCS6B02G365700*	*GIS*	637861404	Seed dormancy, abiotic stress, plant height
Chr7A:85621619	7A	Lat	2.29 × 10^−6^	*TraesCS7A02G133200*	*CML30*	85437099	Root gravitropism
				*TraesCS7A02G133300*		85600437	Female fertility
				*TraesCS7A02G133400* *		85608439	
				*TraesCS7A02G133500* *	*PU1*	85611553	Stripe rust resistance
				*TraesCS7A02G133600*	*DMP10*	85693762	Disease response
				*TraesCS7A02G133700*		85724225	
Chr7A:118327401	7A	Lat	1.63 × 10^−8^	*TraesCS7A02G161500*	*PPR10*	118145757	
				*TraesCS7A02G161600*	*Rf1*	118184773	
				*TraesCS7A02G161900*		118324834	
				*TraesCS7A02G162000*	*CID7*	118326745	Disease response, Seed size
				*TraesCS7A02G162100*		118399509	
				*TraesCS7A02G162200*	*HAL3*	118447874	Grain hardiness, abiotic (salt tolerance), flowering and heading date
				*TraesCS7A02G162400*	*GCR1*	118454859	Disease resistance, seed dormancy cold tolerance
				*TraesCS7A02G162500*		118462549	
Chr7B:559125228	7B	Lat	8.36 × 10^−7^	*TraesCS7B02G313000*		558992275	
				*TraesCS7B02G313100* *		559133986	Drought tolerance, root development, grain density
				*TraesCS7B02G313200* *	*rpl14*	559140536	

^1^ Numbers after the semi-colon indicate the SNP position. * Genes within the same LD block (r = 0.7) of the detected locus. ^2^ Starting position of the gene.

**Table 6 ijms-24-08390-t006:** Genetic differentiation and gene diversity at the significant region on chromosome 4A based on the most significant 5000 SNPs of the genome scan.

Pop	TaClu	TaCul	TaInd	TaLan	TaMac	TaSpe	TaTib	TaYun	TiIsp	TkGeo	TmDom	TmWil	TpXin	TtDom	TtDur	TtKho	TtPer	TtPol	TtRiv	TtWil	TuUra	TvVav
TaClu		0.037	0.110	0.374	0.012	0.001	0.129	0.410	0.079	0.300	0.017	0.016	0.202	0.014	0.034	0.041	0.048	0.043	0.032	0.018	0.016	0.117
TaCul	0.135		0.345	0.000	0.000	0.000	0.350	0.247	0.345	0.247	0.134	0.140	0.300	0.296	0.334	0.326	0.339	0.330	0.318	0.300	0.098	0.338
TaInd	0.089	0.000		0.037	0.000	0.000	0.252	0.379	0.248	0.273	0.303	0.297	0.264	0.307	0.285	0.292	0.289	0.295	0.282	0.307	0.290	0.241
TaLan	0.000	0.101	0.081		0.000	0.000	0.037	0.379	0.011	0.144	0.000	0.000	0.070	0.000	0.001	0.001	0.004	0.002	0.001	0.000	0.000	0.035
TaMac	0.189	0.543	0.533	0.191		0.288	0.000	0.000	0.000	0.000	0.000	0.000	0.000	0.000	0.000	0.000	0.000	0.000	0.000	0.000	0.000	0.000
TaSpe	0.201	0.509	0.485	0.197	0.000		0.000	0.000	0.000	0.000	0.000	0.000	0.000	0.000	0.000	0.000	0.000	0.000	0.000	0.000	0.000	0.000
TaTib	0.082	0.000	0.000	0.079	0.524	0.479		0.385	0.238	0.274	0.300	0.297	0.262	0.307	0.286	0.302	0.300	0.304	0.295	0.309	0.269	0.258
TaYun	0.000	0.026	0.000	0.000	0.351	0.343	0.000		0.309	0.402	0.107	0.101	0.389	0.138	0.213	0.251	0.262	0.243	0.240	0.144	0.100	0.386
TiIsp	0.098	0.002	0.000	0.088	0.524	0.481	0.000	0.008		0.426	0.159	0.161	0.287	0.359	0.329	0.305	0.291	0.297	0.327	0.364	0.138	0.240
TkGeo	0.041	0.000	0.001	0.054	0.514	0.469	0.000	0.000	0.000		0.194	0.191	0.345	0.226	0.260	0.285	0.334	0.284	0.262	0.233	0.193	0.275
TmDom	0.162	0.021	0.022	0.119	0.559	0.522	0.024	0.054	0.025	0.023		0.311	0.246	0.120	0.207	0.190	0.172	0.177	0.215	0.133	0.151	0.304
TmWil	0.163	0.019	0.020	0.120	0.560	0.524	0.022	0.054	0.024	0.020	0.002		0.265	0.102	0.212	0.189	0.179	0.199	0.218	0.112	0.154	0.300
TpXin	0.068	0.000	0.000	0.070	0.522	0.476	0.000	0.000	0.000	0.000	0.022	0.021		0.259	0.251	0.299	0.334	0.302	0.260	0.267	0.253	0.258
TtDom	0.156	0.001	0.000	0.117	0.557	0.524	0.000	0.040	0.002	0.000	0.022	0.021	0.000		0.289	0.335	0.366	0.332	0.293	0.349	0.086	0.307
TtDur	0.143	0.000	0.000	0.107	0.558	0.513	0.003	0.034	0.004	0.003	0.022	0.021	0.002	0.000		0.359	0.349	0.362	0.365	0.300	0.108	0.283
TtKho	0.119	0.001	0.000	0.098	0.536	0.493	0.000	0.020	0.001	0.000	0.023	0.022	0.000	0.001	0.001		0.339	0.339	0.353	0.338	0.184	0.297
TtPer	0.116	0.000	0.000	0.096	0.537	0.493	0.000	0.018	0.001	0.000	0.023	0.021	0.000	0.000	0.001	0.000		0.327	0.334	0.373	0.168	0.293
TtPol	0.121	0.000	0.000	0.098	0.539	0.496	0.000	0.021	0.001	0.000	0.023	0.021	0.000	0.000	0.001	0.000	0.000		0.360	0.357	0.185	0.308
TtRiv	0.126	0.001	0.000	0.101	0.540	0.498	0.000	0.024	0.001	0.000	0.023	0.021	0.000	0.001	0.001	0.000	0.000	0.000		0.304	0.116	0.278
TtWil	0.154	0.001	0.000	0.116	0.555	0.522	0.000	0.039	0.002	0.000	0.022	0.021	0.000	0.000	0.000	0.001	0.000	0.000	0.000		0.088	0.307
TuUra	0.171	0.028	0.031	0.123	0.565	0.528	0.034	0.063	0.035	0.034	0.022	0.020	0.033	0.030	0.031	0.033	0.032	0.032	0.032	0.030		0.266
TvVav	0.086	0.000	0.000	0.080	0.530	0.483	0.000	0.000	0.000	0.000	0.023	0.021	0.000	0.000	0.002	0.000	0.000	0.000	0.000	0.000	0.032	
MSD_intra_	0.04	0.002	4.0 × 10^−5^	0.1	0.114	0.143	8.0 × 10^−4^	0.018	0.001	6.7 × 10^−5^	4.0 × 10^−4^	7 × 10^−4^	0	4.4 × 10^−4^	4.7 × 10^−5^	4.9 × 10^−4^	1.2 × 10^−4^	4.0 × 10^−5^	3.7 × 10^−4^	2.4 × 10^−4^	6.2 × 10^−4^	4.0 × 10^−5^
MSD_inter_	0.12	0.15	4.1 × 10^−5^	0.9	0.13	0.16	8.0 × 10^−4^	0.69	0.001	6.9 × 10^−5^	8.0 × 10^−4^	0.002	0	7.3 × 10^−4^	2.7 × 10^−4^	0.0011	2.5 × 10^−4^	3.2 × 10^−4^	7.8 × 10^−4^	6.8 × 10^−4^	0.001	4.1 × 10^−5^

The numbers above and below the diagonal are the *p*-values and the FST after 10,000 permutations, respectively. The numbers above and below the diagonal are the *p*-values and the FST after 10,000 permutations, respectively. The last two rows indicate the intra-accession and within-population inter-accession gene diversity, respectively. Species: Ta = *T. aestivum*, Ti = *T. ispahanicum*, Tk = *T. karamyschevii*, Tm = *T. monococcum*, Tp = *T. petropavlovskyi*, Tt = *T. turgidum*, Tu = *T. urartu*, Tv = *T. vavilovii*. Wheat types: Clu = Club, Cul = Cultivars, Dom = Domesticated, Dur = Durum, Geo = Georgian, Ind = Indian dwarf, Isp = Ispahanicum, Kho= Khorasan, Lan = Landraces (bread wheat), Mac = Macha, Per = Persian, Pol = Polish, Riv = Rivet, Spe = Spelt, Tib = Tibetan semi-wild, Ura= Urartu, Vav = Vavilovii, Wil = Wild, Xin = Xinjiang, Yun = Yunan.

## Data Availability

The genotype data used in this study were downloaded from https://triticeaetoolbox.org/wheat (accessed on 11 August 2022) and http://bigd.big.ac.cn/gvm (accessed on 29 August 2022). All the data presented in this study are included in the manuscript.

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
