# Peer review of "Historical Selection, Adaptation Signatures, and Ambiguity of Introgressions in Wheat"

_ijms, 2023, doi:10.3390/ijms24098390_

Round 1
Reviewer 1 Report
The presented manuscript outlines the methods and findings from principal component (PC)-based genome scan and genome-environment association (GEA) analyses using an exome capture and whole genome genotyping on two datasets: ~113K SNPs from predominantly bread wheat accessions and ~110K SNPs from ~400 accessions representing all ploidy levels. The study aims to trace historical selection and hybridization signatures. The genome scans on both datasets identified a highly differentiated region on chromosome 4A where accessions in the first dataset were dichotomized into a group (n=691) comprising nearly all cultivars, wild emmer, and most landraces and a second group (n=230) dominated by landraces and spelt accessions. Genome scans and GEA obtained from this study can help to screen germplasm housed in gene banks for breeding and conservation purposes.
It would be helpful for the readers if the authors could reflect on the flow of information captured in the results section matches the introduction section (line # 102-109); in the current text, there is no coherence between this section outlining the key areas of studies and the results sub-sections.
There are minor typos in the text
Author Response
Thank you for your constructive comments and suggestions. They helped us to improve our manuscript.
Please see my replies to your comments in file uploaded here.

Reviewer 2 Report
Dear Author,
Thank you for your submission to IJMS, and for the interesting research presented in "Historical selection, adaptation signatures, and ambiguity of introgressions in wheat." I agree that the authors have done a commendable job of investigating environmental factor-associated loci and genome-environment associations using two SNP datasets, and I appreciate their inclusion of diverse gene pools, which provides a rich evolutionary context.
Regarding the manuscript's strengths, I agree that the population structure is well-suited for the research questions, and the genetic variation is impressively differentiated and widespread across genomic regions.
As for suggested improvements, I think the authors could enhance the manuscript's value by discussing their findings in the context of other species, such as rice, soybean, maize, and so on. Doing so would broaden the potential readership and make the manuscript more appealing to scholars interested in other crops.
Overall, I believe that this manuscript is of high quality and worthy of publication in IJMS after minor revisions, particularly regarding the discussion section. Thank you again for the opportunity to review this submission.
Author Response
Thank you for your constructive comments and suggestions. They helped us to improve our manuscript.
Please see my replies to your comments in the file uploaded here.

Reviewer 3 Report
Sertse et al. conducted genome scans on two SNP datasets to trace wheat's historical selection and hybridization signatures. Environmental factor-associated loci and genome-environment association were also investigated. This is an important study. The manuscript is well-structured and easy to follow. Before the publication of this paper, I would suggest improving the quality of figures to increase readability.
Author Response

(The authors gave the same response as above.)

Reviewer 4 Report
General comments
The manuscript entitled “Historical selection, adaptation signatures, and ambiguity of introgressions in wheat” is good step and facilitate the future studies on what genetics.
Some shortcomings should be resolved before the recommendation of this article to be published.
The manuscript should be checked for minor English and grammar correction.
Abstract
Please add a clear conclusion at the end to this section briefly.
Introduction
This part provide enough information but needs to be re arranged.
Line 41 to line 74 ….these should be summarized in one para and avoid the repetition.
Results
This section is written well.
Discussion
The authors discussed their results in relation with the previous literature, however, justification should be provided for all the results.
Materials and Methods
This section is presented well.
Conclusion
This part is presented well.
Minor English language editing is required.
Author Response

(The authors gave the same response as above.)
